# The economic impact of childhood acute gastroenteritis on Malawian families and the healthcare system

Nathaniel Hendrix,[1,2] Naor Bar-Zeev,[3,4] Deborah Atherly,[1] Jean Chikafa,[4] Hazzie Mvula,[5,6] Richard Wachepa,[4] Amelia C Crampin,[5,6] Themba Mhango,[7] Charles Mwansambo,[8] Robert S Heyderman,[4,9] Neil French,[3,4] Nigel A Cunliffe,[3] Clint Pecenka,[1] for the VacSurv Consortium

## ABSTRACT

**Objectives** This prospective cohort study sought to estimate health system and household costs for episodes of diarrhoeal illness in Malawi.

**Setting** Data were collected in two Malawian settings: a rural health centre in Chilumba and an urban tertiary care hospital in Blantyre.

**Participants** Children under 5 years of age presenting with diarrhoeal disease between 1 January 2013 and 21 November 2014 were eligible for inclusion. Illnesses attributed to other underlying causes were excluded, as were illnesses commencing more than 2 weeks prior to presentation. Complete data were collected on 514 cases at both the time of the initial visit to the participating healthcare facility and 6 weeks after discharge.

**Primary and secondary outcome measures** The primary outcome measure was the total cost of an episode of illness. Costs to the health system were gathered from chart review (drugs and diagnostics) and actual hospital expenditure (staff and facility costs). Household costs, including lost income, were obtained by interview with the parents/guardians of patients.

**Results** Total costs in 2014 US$ for rural inpatient, rural outpatient, urban inpatient and urban outpatient were $65.33, $8.89, $60.23 and $14.51, respectively (excluding lost income). Mean household contributions to these costs were 15.8%, 9.8%, 21.3% and 50.6%.

**Conclusion** This study found significant financial burden from childhood diarrhoeal disease to the healthcare system and to households. The latter face the risk of consequent impoverishment, as the study demonstrates how the costs of seeking treatment bring the income of the majority of families in all income strata below the national poverty line in the month of illness.

For numbered affiliations see end of article.

**Correspondence to**
Dr Clint Pecenka;
cpecenka@path.org

## Strengths and limitations of this study

► We collected costs of illness directly from patient caregivers and clinic records rather than relying on model-derived estimates.
► Collection of information on lost income means that we were able to provide more holistic measures of the effect of illness on household economy.
► Only two clinics were sampled, so the conclusions' generalisability to other locations in Malawi is uncertain.
► Estimates of the contribution of previous visits to overall costs are likely to be confounded by selection bias, as the most severe patients are likely to have been seen before and to have higher costs.

aged 1 to 4 years), and this pathogen causes the largest share of gastroenteritis-related deaths, accounting for 25%.[3 4] The toll of diarrhoeal disease can be mitigated by treatments such as oral rehydration solution, zinc and in some cases, antibiotics or hospital-based care. Vaccines also play an important role in mitigating the spread and severity of diarrhoeal disease. Rotavirus vaccines, for example, have been introduced in many countries, including Malawi, where they prevent infection and severe disease, and have resulted in substantial reductions in rotavirus-associated illness and death.[5 6]

In addition to the burden of disease and death in children, treatment costs can cause significant financial strain to households and healthcare systems alike. In order to understand comprehensively the impact of diarrhoeal disease and the potential economic benefit of vaccines, one must examine the actual costs of care incurred by the healthcare system and by households.

Most economic evaluations have relied on model-derived cost estimates based on extant, retrospective data. Very few studies have

## INTRODUCTION

Diarrhoea is the major cause of childhood illness and death. It is responsible for an estimated 9% of all under-five deaths globally,[1] and in Malawi results in at least 3000 under-five deaths annually.[2] Although the rotavirus vaccine was introduced in 2012, incidence of rotavirus hospitalisation remains high at 123 per 100 000 infants (60 per 100 000 children

prospectively collected direct individual patient-level costs of diarrhoeal illness at both inpatient and outpatient settings.[7] Even fewer have collected such data from health facilities and households, including direct and indirect medical costs and income loss estimates.[8] Additionally, many studies lack data on household income that would permit examination of differential short-term financial effects of illness by income level.

In this post hoc analysis of data collected for a cost-effectiveness analysis,[9] we sought to provide insight into the cost of treating diarrhoeal disease from the health system and societal perspectives in the context of a low-income country where receipt of both doses of the rotavirus vaccine is near 90%.[10] In other low-income settings, geographic setting (urban/rural), treatment setting (inpatient/outpatient), household income and duration of illness have consistently demonstrated correlation with the economic burden of diarrhoeal disease.[7 11 12] We sought to explore these factors that drive cost of illness in this sub-Saharan, low-income setting, as well as to leverage our capture of prior costs to examine the effects of multiple sessions of care on cost.

## CONTEXT AND METHODS
### Study area and population
This prospective cohort study was conducted in Malawi, a country of approximately 16 million people in southern Africa. Its per capita gross national income of US$350 places it 210 of 216 countries and territories ranked by the World Bank.[13] Over 80% of its population is rural, with much of that population engaged in subsistence farming.[14]

The study was conducted at two sites: Queen Elizabeth Central Hospital (QECH), a national referral hospital that provides both inpatient and outpatient care, and Chilumba Rural Hospital (CRH), which provides outpatient primary care for children under 5 years and accepts inpatients, although generally low acuity cases. QECH is located in Blantyre, a district of approximately 1.3 million in Malawi's Southern Region. It receives approximately 90 000 presentations per year, 25 000 of which are admitted to the inpatient facility. CRH is located in Karonga, a district of approximately 200 000 in the Northern Region. Chilumba hosts a demographic surveillance site with a population approaching 40 000.[15] It sees 7500 outpatients per year, of which about 700 are admitted inpatient. Both sites provide basic inpatient and outpatient care free of charge and were purposively selected to be representative of the care provided at different levels of the health system.[9]

### Inclusion criteria
Eligible participants were children under 5 years of age living in Blantyre district or within the demographic surveillance population in Chilumba and presenting with diarrhoea at either facility during the study period (1 January 2013 to 21 November 2014). Illnesses

attributed to other underlying causes were excluded, as were illnesses commencing more than 2 weeks prior to presentation. In addition to identifying participants at registration/triage, admission logs and electronic records (at QECH only) were reviewed for missing cases.

### Data collection
After identifying each participant and obtaining consent from their caregiver, a trained research assistant administered a standard case report form to the caregiver. The questionnaire included demographic information, socioeconomic status and the history of the child's illness prior to presentation at the facility, as well as all illness-associated costs incurred prior to presenting at the facility. Daily review of medical charts was undertaken during admission for inpatients to gather information on drugs dispensed, and laboratory investigations or procedures performed. All participants' homes were visited by research staff 6 weeks postdischarge to evaluate whether any subsequent illness-related costs were incurred by the household.

### Categorisation of expenses
Pretreatment costs included direct medical costs paid by the household for care sought prior to the date of the index visit, which we defined as the date of an outpatient visit at either facility or the first day of an inpatient stay. Since the health facilities at which previous care was sought were not study sites, costs to the health system were not included in the pre-index costs.

Costs of the index visit were divided into health system and household costs. Costs to the health system included drugs, laboratory investigations, staff and facility costs such as laundry, kitchen, sanitation and security. The cost of drugs was calculated based on acquisition costs for drugs used at the index visit, with the quantity of injectable drugs being rounded up to the nearest full phial size. Investigations performed were costed on charge per test, except for those performed as part of clinical research, which were excluded. Staff costs for inpatient visits used the combined daily salary of all staff present divided by the number of beds in the ward, then multiplied by the patient's length of stay in days; staff costs of emergency outpatient visits were calculated by dividing the combined daily salary of staff by the mean daily number of visits to the department. Facility costs were calculated by dividing the facility's mean daily cost for services and subsistence goods by the number of hospital beds, then multiplying by the patient's length of stay. The cost of consumable goods in the clinic such as tubing for intravenous drips was not included. Household costs included transportation to the index visit for the patient, their main caretaker and any other household visitors during admission; direct medical costs (costs of consultations, drugs and diagnostics not covered by the essential health package); lost income attributable to the episode of disease; and subsistence (food and shelter) during the visit. Lost income was calculated by dividing the self-reported monthly income

**Table 1** Patient characteristics

| | Rural | | Urban | | |
| | Inpatient | Outpatient | Inpatient | Outpatient | Total |
|---|---|---|---|---|---|
| N (% of total) | 22 (4%) | 108 (21%) | 269 (52%) | 120 (23%) | 519 |
| Mean Vesikari score (SD) | 6.0 (2.9) | 2.6 (2.2) | 11.1 (3.9) | 2.0 (3.4) | 7.0 (5.5) |
| Sex: male, not recorded(%) | 13 (59%), 0 | 67 (62%), 2 (2%) | 159 (59%), 2 (1%) | 62 (52%), 1 (1%) | 301 (58%), 5 (1%) |
| Mean age, months (SD) | 14.0 (12.7) | 15.8 (11.3) | 13.4 (7.4) | 15.8 (11.8) | 14.3 (9.3) |
| Median length of stay, days (IQR) | 2 (2.5) | N/A | 3 (2) | N/A | 3 (2) (inpatients only) |

by the number of days in the month, then multiplying by the number of days spent in care taking for this episode of illness.

Ongoing costs were assessed at a follow-up visit to the patient's home 6 weeks after facility discharge. Costs related to any long-term disability caused by the episode were not included among these ongoing costs. Again, costs to the health system for further treatment could not be ascertained due to other sites' being outside the network of research sites.

All costs were collected in Malawi kwacha and converted to US$ based on the mid-market exchange rate per the Malawi Reserve Bank on 15 July 2014.

### Statistical analysis

The study's minimum sample size of 88 was calculated to provide an estimate of household costs with a margin of error ≤10% given a coefficient of variation of 0.5.[9 16]

Income quintiles were calculated based on self-reported monthly income within the sample itself—that is to say that a household labelled as belonging in the highest income quintile is within the top 20% of household incomes in this sample. Using one-way analysis of variance (ANOVA), we tested for differences among income quintiles in mean overall household costs at each geographic and treatment setting.

To explore associations proposed in previous research between prior treatment and cost, we constructed a generalised linear model of total costs and household costs with log link on gamma distributional family, adjusted for length of stay, geographic setting and treatment setting. We used a $X^2$ test to compare the proportion of patients with prior/ongoing costs in different settings, and a rank-sum test to assess differences in component

costs between these groups. We also used rank-sum tests to assess the significance of differences in total (health system plus household) and household cost between geographic and treatment settings.

### Ethics

Ethical approval was provided by the National Health Sciences Research Committee, Lilongwe, Malawi (1073), and by the Research Ethics Committee of the University of Liverpool, UK (000490). Written consent was obtained from the parents or guardians of participating children.

### RESULTS
### Participants

A total of 529 individuals participated in this study, although 15 were excluded due to incomplete data (see table 1). The majority (389) were recruited at QECH in Blantyre and the remaining 130 participants were recruited at CRH in Karonga. Inpatient cases accounted for 69% (269/389) of the cases at QECH while only 17% (22/130) of cases in Karonga were admitted, reflecting a more serious case mix at a much larger national referral level hospital in Blantyre.

Total costs for the episode of illness and costs by payer are reported in table 2, sorted by geography and treatment setting. The least expensive visit type was a rural, outpatient visit with total costs averaging $13.57; rural, inpatient visits were the costliest, averaging $76.94. Rank-sum tests confirmed the significance of differences in both total and household costs between geographic and treatment settings (p<0.001 in all cases). Across all settings, the health system bore an average of 76% of the costs for each episode of illness.

**Table 2** Mean (SD) total direct and indirect costs for the episode of illness by geography and treatment setting

| | Rural | | Urban | |
| | Inpatient | Outpatient | Inpatient | Outpatient |
|---|---|---|---|---|
| Health system costs | $55.04 ($26.82) | $8.02 ($2.90) | $47.21 ($56.69) | $7.18 ($4.05) |
| Household costs | $19.16 ($19.30) | $1.81 ($2.58) | $25.36 ($20.47) | $15.48 ($20.75) |
| Total costs | $76.94 ($32.61) | $13.57 ($4.70) | $73.78 ($51.39) | $23.13 ($21.68) |

**Table 3** Mean (SD) costs to health system for index visit by category

|  | Rural | | Urban | |
|---|---|---|---|---|
|  | **Inpatient** | **Outpatient** | **Inpatient** | **Outpatient** |
| Drugs | $4.46 ($3.94) | $1.45 ($2.53) | $9.40 ($20.29) | $1.19 ($2.16) |
| Labs | $5.39 ($1.07) | $4.96 ($1.17) | $7.60 ($5.21) | $4.73 ($3.36) |
| Staff | $17.03 ($10.37) | $1.61 ($0) | $21.47 ($30.51) | $1.26 ($0) |
| Facilities | $28.35 ($17.26) | $0 ($0) | $8.73 ($12.41) | $0 ($0) |
| Total | $55.04 ($26.82) | $8.02 ($2.90) | $47.21 ($56.69) | $7.18 ($4.05) |

We performed a sensitivity analysis to assess the potential impact to overall costs at the index visit if the 15 dropped observations were included. Were these costs representative of the 25th percentile of costs, mean overall costs would decrease from $39.15 to $38.32; were they in the 75th percentile of costs, mean overall costs would rise to $39.44. This would result in a 2% decrease in total costs, or a 0.6% increase, respectively, compared with the weighted average of costs across all visit types.

For inpatient settings, staff and facility costs contribute the most to health system costs (see table 3). These two categories represent between 60% and 80% of total costs across geographic settings. Laboratory investigations dominate outpatient costs, representing roughly 60% in both urban and rural outpatient settings.

Lost income and direct medical expenses make up the majority of household costs for all except rural, outpatient visits, whose costs are dominated by transportation and lost income (see table 4). Of patients seen at the urban facility, 35% had sought treatment prior to their index visit, compared with 2% of patients seen at the rural facility (p<0.001). On the other hand, 13% of patients in Karonga had ongoing costs after the index visit, compared with 2% of patients seen at the Blantyre site (p<0.001).

Using one-way ANOVA to detect differences in mean household expenditure (exclusive of lost income) between income quintiles, we found a significant difference only in urban, inpatient encounters (see figure 1).

The highest income level in this group (>$79.50 per month) had higher expenditures than any other income group, with the difference between this group and the lowest income group (<$19.32 per month) being especially noticeable. The ANOVA test showed no significant differences for any other combination of geographic and treatment settings.

Costs to the households of patients for whom previous treatment was sought were 2.04 times higher (p<0.001) once adjusted for length of stay, geography and treatment setting; overall costs were 1.25 times higher (p=0.016). This difference in overall costs is driven in part by spending prior to the index visit, which averaged between $0.92 for rural, outpatient index visits and $3.01 for rural, inpatient index visits. A rank-sum test on transportation costs by prior treatment status showed significantly higher transportation costs for this group (p<0.001), suggesting the source of these differences. Median monthly income for the group who sought prior treatment was higher than the group who did not, at $34.09 and $22.73, respectively. A rank-sum test again showed a significant difference in both monthly income and lost income due to the illness (length of stay was not significantly different between the groups). We repeated the rank-sum test looking for variation in transportation costs during the rainy seasons (November through April) of those who received prior treatment, but this test's results were not significant.

**Table 4** Mean (SD) household costs for an episode of illness

|  | Rural | | Urban | | |
|---|---|---|---|---|---|
|  | **Inpatient** | **Outpatient** | **Inpatient** | **Outpatient** | **Total** |
| Direct medical expenses prior to index visit | $0.21 ($0.72) | $0.08 ($0.58) | $0.98 ($3.92) | $0.71 ($1.55) | $0.77 ($3.13) |
| Direct medical expenses of index visit | $8.61 ($9.62) | $0.24 ($0.68) | $10.69 ($10.66) | $5.85 ($10.21) | $7.31 ($10.17) |
| Transportation to and subsistence at index visit | $4.38 ($5.28) | $0.51 ($0.54) | $2.68 ($3.00) | $0.45 ($1.91) | $2.04 ($2.83) |
| Lost income due to index visit | $4.13 ($9.62) | $0.50 ($0.68) | $8.06 ($10.66) | $6.22 ($10.22) | $6.36 ($10.17) |
| Direct medical expenses after index visit | $0.87 ($2.10) | $0.38 ($1.66) | $0.21 ($2.41) | $0.16 ($1.19) | $0.26 ($2.02) |
| Total household costs | $19.16 ($19.30) | $1.81 ($2.58) | $25.36 ($20.47) | $15.48 ($20.75) | $19.32 ($20.66) |

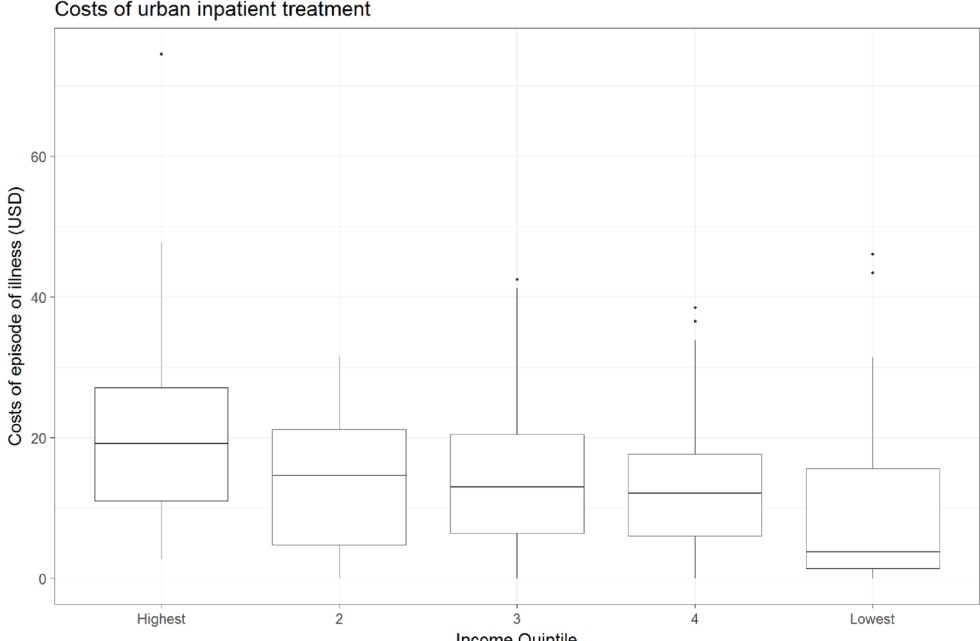

**Figure 1** Mean total household cost (not including lost income) of inpatient, urban index visit by income.

## DISCUSSION

Even following the introduction of the rotavirus vaccine, diarrhoeal disease continues to be a significant burden on the world and on low-income countries in particular. While deaths among African children due to diarrhoea declined 43% between 2000 and 2013,[17] the number of cases remains high, with one study estimating 3.3 episodes of diarrhoea annually per child in Africa.[18]

This study focused on the economic burden of these diseases by combining costs ascertained through both interviews and chart review. It found significant household costs primarily in the inpatient and urban settings with lost income and direct medical costs dominating the contributing costs. Even rural, outpatient visits can be financially problematic, however, if they are frequent, or if the family is especially vulnerable. Household costs for the episode of illness (excluding lost income) exceeded monthly household income in 8% of cases treated inpatient and 3% of cases treated outpatient; including lost income, these numbers rise to 17% and 9%, respectively.

Taking into account lost income, mean household costs for patients seen in the rural health centre were $19.16 and $1.81 for inpatient and outpatient treatment, while patients seen at the urban hospital had mean household costs of $25.36 and $15.48 for inpatient and outpatient treatment. The high cost of inpatient treatment in the urban setting likely reflects the higher acuity of illness at that setting: the mean Vesikari score of study participants seen in QECH's inpatient ward was 11, which indicates severe disease. Rural inpatients, by comparison, had a mean Vesikari score of 6.

Putting household expenditures in context, 72.2% of the Malawian population lives on <$1.25 per person per day.[19] Even an outpatient visit in a rural setting, then, is likely to impose costs exceeding a day's income for this

population, while a costlier inpatient visit is likely to account for almost 3 weeks' income. The equivalents for a US household would be to face costs of $147 and $3,095, respectively.[20] It should be noted, though, that in a low-income setting, a much larger percentage of income goes to core goods such as food and shelter: as such, it is more difficult to offset health-related costs. Having savings adequate to cover medical costs is also less common in this setting, where, although 76% report saving when possible, 90% of households say that they sometimes run out of money for basic goods.[21] This can bring about debt and reduced expenditure on clothing, education, shelter and other similar goods.

The results of our study can be compared with several other studies of the economic burden of diarrhoeal disease in the low-income, sub-Saharan context. An analysis of economic data from the Global Multicenter Enteric Study found mean household costs of $2.77, $6.57 and $4.33 (in 2014 USD) for Gambia, Kenya and Mali, respectively, for a representative sample of inpatient and outpatient medical encounters, not including lost income.[22] A Rwandan study of inpatient admissions for childhood gastroenteritis found household costs of $66.08 (including lost income) and costs to the medical system of $34.83.[23] Finally, a study from Ghana found costs to the medical system of $5.14 for outpatient visits and $122.07 for inpatient visits.[8]

When costs are analysed by income quintile, mean household expenditure by quintile is only significantly different in the urban, inpatient setting. This demonstrates that the poorest segments of Malawian society who attend public healthcare facilities generally incur the same costs as their wealthier peers in the same settings. This finding is supported by other evidence from Malawi which found that care-seeking behaviour

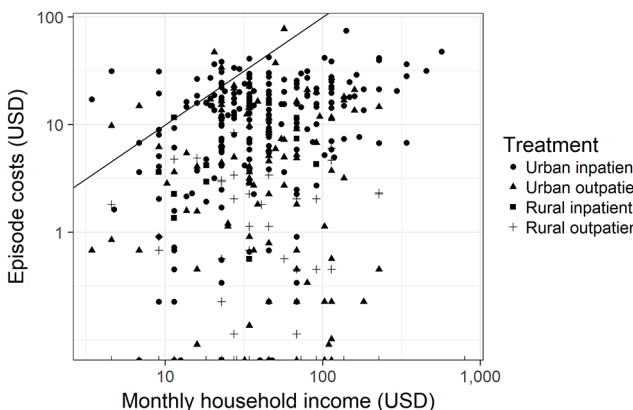

**Figure 2** Log monthly household income versus log costs for the episode of illness, not including lost income. The line represents episode costs equal to monthly income.

for children with diarrhoea does not vary by income level.[24]

The pattern of identical expenditure across income quintiles in most settings means that the poorest households expose themselves to greater risk of impoverishment by seeking treatment (see figure 2). After accounting for the costs of the episode of illness, no household in the lowest income quintile of our sample was left with more than $0.60 of income per day for the month in which the illness occurred—well below the government's definition of 'ultrapoverty' as $1.74 per household per day.[25] Furthermore, the costs of the episode of illness exceeded the monthly income of 27% of households in this group. Even among the highest income quintile of our sample, 19% of households were brought temporarily below the poverty line of $2.80 per day by the costs of the episode of illness.

We are left to explain why spending increases with income quintile in the urban inpatient setting (see figure 1). One explanation may be that price sensitivity may play a role in care decisions in the urban inpatient setting. If this were the case, the children of poorer Malawians may not receive the full extent of care that they need due to concerns about the costs associated with seeking treatment, even when the treatment itself is free. Mean total household costs were the highest for this setting, and the patients' caretakers must balance the risks of expenses that they cannot afford with the risk that their child's illness will worsen when they choose to seek or not seek care. Lower income families may, rationally, apply a different threshold for seeking treatment than higher income families. Another possibility may be a tendency of higher income families to bring more visitors or to buy higher cost subsistence goods. In order to test this possibility, we ran an ANOVA test on mean spending by income quintile on visitors, transportation and shelter costs (both together and separately), but did not find significant differences. Instead, the same test shows significantly increased mean spending on direct medical expenses as income quintile increases.

One driver of high overall costs in the urban setting at large may be the seeking of prior care. Of patients

seen in the urban setting, 35% had been seen at a prior healthcare facility. Testing the relationship between prior care and income quintile using a logistic regression does not suggest a significant relationship between the two, whether income quintile is represented as a continuous or factor variable. Excluding lost wages, household costs for cases in which prior care is sought cost twice as much as cases in which the index visit is the first encounter with the healthcare system during this episode of illness. Transportation costs are the likely source of difference, suggesting that patients who sought treatment prior to the index visit come to the hospital either from farther away or more urgently than those for whom the index visit was their first visit related to this episode. Transportation costs predominate costs accrued prior to the index visit as well, accounting for 62%; spending at health centres and pharmacies accounted for 13% and 11%, respectively. Selection bias likely plays a role as well, as the most seriously ill patients are likely both to be seen multiple times and to incur greater overall costs.

The conclusions in this study are limited by its being a post hoc, unpowered analysis of data gathered for a cost-effectiveness study. That is, the sample size of the study was not powered to detect differences between treatment settings, geography or any of the other characteristics examined in our regression analysis. The study's generalisability may also be limited due to the data collection's having taken place in two locations that represent extremes of acuity and population density within the Malawian healthcare system. We cannot rule out greater geographic variation in financial outcomes than our study results suggest.

Future work in this area should focus on the long-term consequences to the household of financial shock associated with an episode of diarrhoeal disease, or with repeated diarrhoea due to undernourishment and other risk factors. Especially of interest is the long-term loss of economic potential due to childhood illness: if a child is frequently or severely ill, health-related expenditures, including time needed for caretaking, may preclude investment in other areas that could improve the financial situation of a household.

## CONCLUSION

This study has evaluated directly the cost of 514 episodes of childhood diarrhoeal disease to both the households and health system of Malawi. It found that households bear a substantial fraction of the total costs of an episode of illness and that there is a considerable risk of impoverishment due to severe or repeated episodes of diarrhoeal disease.

The conclusions of this study are broadly applicable in the sub-Saharan setting—especially to those countries where the rotavirus vaccine is widely used—regardless of whether healthcare is provided to the public for free or through an insurance programme. In the latter case, only direct medical costs to the household would be covered. Lost income, transportation and ongoing costs of care were all found in this study to impose substantial financial

burdens, and this observation would not be changed by the use of health insurance.

This study's findings can be applied to cost-effectiveness analyses of enteric vaccines and other means of preventing the transmission of diarrhoeal disease and can be used by healthcare planners to help prioritise interventions. By providing data on the lost financial opportunity caused by an episode of diarrhoeal disease, this study allows for a more precise accounting of how an averted episode of diarrhoeal disease should be valued.

**Author affiliations**
[1]Center for Vaccine Innovation and Access, PATH, Seattle, Washington, USA
[2]Pharmaceutical Outcomes Research and Policy Program, University of Washington, Seattle, Washington, USA
[3]The Centre for Global Vaccine Research, Institute of Infection & Global Health, University of Liverpool, Liverpool, Merseyside, UK
[4]Malawi-Liverpool-Wellcome Trust Clinical Research Programme, College of Medicine, University of Malawi, Blantyre, Malawi
[5]Karonga Prevention Study, Chilumba, Malawi
[6]London School of Hygiene and Tropical Medicine, London, UK
[7]Queen Elizabeth Central Hospital, Blantyre, Malawi
[8]Ministry of Health, Lilongwe, Malawi
[9]Division of Infection & Immunity, University College London, London, UK

**Collaborators** James Beard (London School of Hygiene and Tropical Medicine, London, UK); Anthony Costello (WHO, Geneva, Switzerland; formerly University College London (UCL), London, UK); Miren Iturriza-Gomara (University of Liverpool (UoL), Liverpool, UK); Khuzwayo Jere (UoL) Carina King (UCL, London, UK); Sonia Lewycka (University of Auckland, Auckland, New Zealand; formerly UCL); Osamu Nakagomi (Nagasaki University, Japan); Umesh Parashar (Centers for Disease Control & Prevention (CDC), Atlanta, GA, USA); Tambosi Phiri (Mai Mwana Project, Mchinji, Malawi); Jacqueline E Tate (CDC); Jennifer R Verani (CDC); Cynthia G Whitney (CDC).

**Contributors** NB-Z, DA and NAC designed the study with input from ACC, CM, RSH and NF. JC led data collection efforts. HM and TM contributed to data collection. NH, RW, NB-Z, CP and DA designed and performed the data analysis. NH wrote the first draft of the manuscript. All listed authors contributed to the interpretation of the data, contributed to the writing of the manuscript, reviewed the work prior to submission and have agreed to be responsible for its content.

**Funding** This work was supported by Wellcome Trust Programme Grant (WT091909), PATH and the Bill & Melinda Gates Foundation (OPP1053539).

**Competing interests** NB-Z, NAC and NF have received investigator-initiated research grants from GSK, and NB-Z and NAC from Takeda Pharmaceuticals. All other authors have no conflicts to declare.

**Patient consent** No identifiable medical information is included in this manuscript.

**Ethics approval** National Health Sciences Research Committee, Lilongwe, Malawi (1073), and by the Research Ethics Committee of the University of Liverpool, UK (000490).

**Provenance and peer review** Not commissioned; externally peer reviewed.

**Data sharing statement** Data collected as part of this study is not available due to its not having been included in the approved protocol.

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
