## [Reviewer comments · BMJ Open]

ARTICLE DETAILS

TITLE (PROVISIONAL)	The economic impact of childhood acute gastroenteritis on Malawian families and the healthcare system: a prospective cohort study
AUTHORS	Hendrix, Nathaniel; Bar-Zeev, Naor; Atherly, Deborah; Chikafa, Jean; Mvula, Hazzie; Wachepa, Richard; Crampin, Amelia; Mhango, Themba; Mwansambo, Charles; Heyderman, Robert; French, Neil; Cunliffe, Nigel; Pecenka, Clint

VERSION 1 - REVIEW

REVIEWER	Chandrashekhar T Sreeramareddy Department of Community Medicine, International Medical University, Bukit Jalil, Kuala Lumpur, Malaysia
REVIEW RETURNED	28-Apr-2017

GENERAL COMMENTS	This study though has some limitations of the small sample, has novelty, has potential to advance economic evaluation of diarrhea prevention. There are a few minor issues which are included as comments in the file uploaded here. They are use of terminology, sequence of writing discussion sections etc The reviewer also provided a marked copy with additional comments. Please contact the publisher for full details.
--

REVIEWER	Birger C Forsberg Dept or Public Health Sciences Karolinska Institutet Stockholm Sweden
REVIEW RETURNED	16-May-2017

GENERAL COMMENTS	Excellent article which is thorough and free of errors. I could not find any.
---

REVIEWER	Tharani Loganathan Ministry of Health, Malaysia
REVIEW RETURNED	28-May-2017

GENERAL COMMENTS	This paper is interesting examination of costs incurred by households and the health care system for the management of childhood diarrhoea at two healthcare facilities in Malawi. It is clearly written and to the point.
--

	Here are a few minor comments:  1. The authors mention that rotavirus infection is the most common cause of childhood diarrhoeal deaths (25%) in Malawi. Rotavirus vaccine has been introduced in Malawi. It would be useful to note when rotavirus vaccines were introduced and the current incidence of rotavirus gastroenteritis in Malawi. 2. Since this study examines household costs of childhood diarrhoea, it would be more interesting to know more about the healthcare system and health financing in Malawi.  a. We are told that both study sites provide inpatient and outpatient care free of charge, however direct medical expenses of the index visits are not explained. Are there a consultation or medication fee? Were the study sites public or private facilities? b. Also, the authors should describe direct medical expenses prior to index visit to the hospital. Do the patients' access care at public or private facilities, traditional medication practitioners, or pharmacies?
--	---

VERSION 1 – AUTHOR RESPONSE

We would like to thank the three reviewers for their consideration of this paper's contents.

We will respond to individual points below:

> There are a few minor issues which are included as comments in the file uploaded here.

The paper attached to this review was titled "Country characteristics and variation in diabetes prevalence among Asian countries." We would appreciate seeing the comments this reviewer provided on our paper and will consider the revisions contained within it when the correct file is uploaded.

> It would be useful to note when rotavirus vaccines were introduced and the current incidence of rotavirus gastroenteritis in Malawi.

We have added this information to line 68 of the revised manuscript.

> We are told that both study sites provide inpatient and outpatient care free of charge, however direct medical expenses of the index visits are not explained.

This information has been added to line 147 of the revised manuscript.

> Also, the authors should describe direct medical expenses prior to index visit to the hospital.

This information was added to line 312 of the revised manuscript.

VERSION 2 – REVIEW

REVIEWER	Chandrashekhar T Sreeramareddy International Medical University Malaysia
REVIEW RETURNED	01-Jul-2017

GENERAL COMMENTS	Thank you for the revised submission. I cannot remember if there were major issues on the first version or i could see my own or other reviewers' comments. On re-reading the manuscript, authors
---

	highlights some minor revision made. I may have commented directly in the MS PDF file. Nevertheless, overall, the MS is in an acceptable standard. It would be good to append all reviewers' comments when a revised version is resent to reviewers.
--	--

REVIEWER	Tharani Loganathan Ministry of Health, Malaysia
REVIEW RETURNED	04-Jul-2017

GENERAL COMMENTS	The authors have addressed my queries adequately.
---